# A Comparison between SARS-CoV-2 and Gram-Negative Bacteria-Induced Hyperinflammation and Sepsis

**DOI:** 10.3390/ijms242015169

**Published:** 2023-10-14

**Authors:** Klaus Brandenburg, Raquel Ferrer-Espada, Guillermo Martinez-de-Tejada, Christian Nehls, Satoshi Fukuoka, Karl Mauss, Günther Weindl, Patrick Garidel

**Affiliations:** 1Brandenburg Antiinfektiva, c/o Forschungszentrum Borstel, Leibniz-Lungenzentrum, Parkallee 10, 23845 Borstel, Germany; kbrandenburg@fz-borstel.de (K.B.); karlmauss@googlemail.com (K.M.); 2Department of Microbiology, University of Navarra, IdiSNA (Navarra Institute for Health Research), Irunlarrea 1, E-31008 Pamplona, Spain; gmartinez@unav.es; 3Department of Systems Biology, Harvard Medical School, Boston, MA 02115, USA; 4Forschungszentrum Borstel, FG Biophysik, Parkallee 10, 23845 Borstel, Germany; cnehls@fz-borstel.de; 5National Institute of Advanced Industrial Science and Technology (AIST), Takamatsu 761-0395, Japan; sfukuoka.pc991@gmail.com; 6Sylter Klinik Karl Mauss, Dr.-Nicolas-Strasse 3, 25980 Westerland (Sylt), Germany; 7Pharmazeutisches Institut, Abteilung Pharmakologie und Toxikologie, Universität Bonn, Gerhard-Domagk-Str. 3, 53121 Bonn, Germany; guenther.weindl@uni-bonn.de; 8Physikalische Chemie, Martin-Luther-Universität Halle-Wittenberg, 06108 Halle (Saale), Germany

**Keywords:** sepsis, lipopolysaccharide, Gram-negative bacteria, COVID-19 pandemic, hyperinflammation, TLR4, cytokines, ARDS, Aspidasept

## Abstract

Sepsis is a life-threatening condition caused by the body’s overwhelming response to an infection, such as pneumonia or urinary tract infection. It occurs when the immune system releases cytokines into the bloodstream, triggering widespread inflammation. If not treated, it can lead to organ failure and death. Unfortunately, sepsis has a high mortality rate, with studies reporting rates ranging from 20% to over 50%, depending on the severity and promptness of treatment. According to the World Health Organization (WHO), the annual death toll in the world is about 11 million. One of the main toxins responsible for inflammation induction are lipopolysaccharides (LPS, endotoxin) from Gram-negative bacteria, which rank among the most potent immunostimulants found in nature. Antibiotics are consistently prescribed as a part of anti-sepsis-therapy. However, antibiotic therapy (i) is increasingly ineffective due to resistance development and (ii) most antibiotics are unable to bind and neutralize LPS, a prerequisite to inhibit the interaction of endotoxin with its cellular receptor complex, namely Toll-like receptor 4 (TLR4)/MD-2, responsible for the intracellular cascade leading to pro-inflammatory cytokine secretion. The pandemic virus SARS-CoV-2 has infected hundreds of millions of humans worldwide since its emergence in 2019. The COVID-19 (Coronavirus disease-19) caused by this virus is associated with high lethality, particularly for elderly and immunocompromised people. As of August 2023, nearly 7 million deaths were reported worldwide due to this disease. According to some reported studies, upregulation of TLR4 and the subsequent inflammatory signaling detected in COVID-19 patients “mimics bacterial sepsis”. Furthermore, the immune response to SARS-CoV-2 was described by others as “mirror image of sepsis”. Similarly, the cytokine profile in sera from severe COVID-19 patients was very similar to those suffering from the acute respiratory distress syndrome (ARDS) and sepsis. Finally, the severe COVID-19 infection is frequently accompanied by bacterial co-infections, as well as by the presence of significant LPS concentrations. In the present review, we will analyze similarities and differences between COVID-19 and sepsis at the pathophysiological, epidemiological, and molecular levels.

## 1. Risk Factors and Complications of COVID-19

The Coronavirus disease of 2019 (COVID-19) due to the coronavirus SARS-CoV-2, is a pandemic with a high rate of mortality [1,2]. The first cases were reported at the end of 2019 in Wuhan, China, and were diagnosed with severe acute respiratory syndrome (SARS) leading to a potentially life-threatening disease. The symptoms of this pathological condition are fever, shortness of breath, cough, and sudden onset of anosmia (“smell blindness”), ageusia (loss of the sense of taste), or dysgeusia (disorder of the sense of taste). In most cases, approximately 80%, COVID-19 is mild or moderate, but it can evolve into severe or critical clinical presentations with a significant risk of mortality in about 14% and 5% of the cases, respectively [3].

The causative agent of COVID-19, severe acute respiratory syndrome coronavirus-2 (SARS-CoV-2), is an enveloped positive single-stranded RNA virus, with a genome 8.4–12 kDa in size. The 5′ terminal part of this genome, which is rich in open reading frames, encodes proteins essential for virus replication. On the other hand, the 3′ terminal includes five structural proteins, spike protein (S), responsible for the pathogenesis in the human species; the membrane protein (M); nucleocapsid protein (N); envelope protein (E); and hemagglutinin-esterase protein (HE) [4].

Numerous studies have analyzed which comorbidities are more commonly associated with COVID-19 severity [5,6,7,8,9]. Interestingly, all these meta-analyses consistently showed that patients suffering from diabetes, hypertension, cancer, and cardiovascular diseases were at higher risk of developing severe COVID-19. Association between other comorbidities and disease burden was also strong, although their relative contribution to disease severity varied among the distinct meta-analyses (Table 1).

However, to date, the decisive pathophysiologic processes that are responsible for COVID-19 patient morbidity and mortality remain unclear. Chen et al. reported that acute respiratory distress syndrome (ARDS), respiratory failure, multiple organ dysfunction syndrome (MODS) and septic shock were complications strongly associated with critical cases of coronavirus disease [5]. This meta-analysis was particularly relevant as it examined data from 187 studies describing 77,013 patients [5]. Other studies reached the same conclusions [10,11,12]. Importantly, severe cases of non-COVID-19 sepsis caused by respiratory pathogens lead to complications similar to those described by these authors, thereby suggesting that COVID-19 mortality may be the result of sepsis. To address this hypothesis, Ahlström et al. compared the impact of comorbidities on mortality in patients with COVID-19 and sepsis [8]. These authors reported that mortality was significantly reduced in the COVID-19 patients compared with those with sepsis, whereas the use of invasive mechanical ventilation was more common in COVID-19 than in sepsis patients. However, the key conclusion of this study is that almost all the investigated comorbidities were shared between COVID-19 and sepsis patients. Consistent with this finding, sepsis scores have been consistently shown to predict COVID-19 outcomes including death, ICU (intensive care unit) transfer, and respiratory failure [13,14]. For example, 78% of severely ill COVID-19 patients met the Sepsis 3.0 criteria for sepsis/septic shock with ARDS being the most common organ dysfunction at 88% [15].

## 2. COVID-19-Induced Sepsis, Immunotherapies, and Antiviral Treatments

During COVID-19 disease, both the innate and the adaptive immune responses experience dysregulation. The first clinical reports from early 2020 highlighted high plasma levels of interleukin-6 (IL-6), IL-1, tumor necrosis factor α (TNF-α), ferritin, and increased amounts of other inflammatory biomarkers. This underlined the assumption that COVID-19 was comparable to sepsis and led to the idea that these biomarker levels were the cause for organ failure and, thus, needed to be suppressed [16,17,18,19,20,21]. Therefore, several clinical trials started using anti-inflammatory therapies to try to reduce the cytokine plasma levels [22,23,24] (Table 2). These clinical trials have not been successful so far and, in some cases, have even worsened patient outcomes [25,26].

Currently it is understood that, for instance, early conclusions based on IL-6 concentration were not robust as predictors of COVID-19 prognosis. Although initial data showed abnormally elevated IL-6 concentrations in COVID-19 patients of a few hundred pg/µL, these levels were modest compared with those measured in septic shock patients undergoing a cytokine storm. Specifically, the levels measured in the plasma of the latter patients exceeded those of COVID-19 patients by a factor of 10–20, leading to IL-6 plasma concentrations of thousands of pg/µL. In addition, it was soon observed that elevation of IL-6 levels associated with COVID-19 was a transient phenomenon. Thus, Gu et al. (2020) showed that wild-type and ACE2-expressing (adenovirus-5/human angiotensin-converting enzyme 2) BALB/c mice challenged with a combination of polyinosinic-polycytidylic acid (an immunostimulant used in the form of its sodium salt to simulate viral infections) [27] and a recombinant SARS-CoV-2 spike-extracellular domain protein expressed high levels of TNF-α and underwent 100-fold increases in IL-6 at 6 h post-challenge. However, the levels of TNF-α and IL-6 later returned to normal ranges from the bronchoalveolar lavage fluid (BALF) after 24 h of the exposure [28]. As a result of these studies, our current knowledge about the disease evolution considers not only the plasma concentrations of inflammatory markers, but also the phase of the disease (Table 3).

According to several studies, the inflammatory phase for patients with severe COVID-19 is limited to the initial period of the disease [28]. The subsequent chronic basal inflammation, which lasts several days, leads the immune system towards a refractory state, which is also observed in protracted sepsis. A comparative study of patients with severe and mild COVID-19 concluded that all cytokines, except IL-6 and IL-10, reached their peak level in serum 3–6 days after the beginning of the disease. IL-6 levels on the other hand, began to drop approximately 16 days later, and IL-10 levels were at their lowest 13 days after disease onset. Interestingly, the cytokine levels reached similar points for all patients with severe and mild disease 16 days after disease onset. This observation corresponds to the most advanced phases of sepsis, in which the macrophages develop a refractory state characterized by a strong inhibition of the NF-κB and interferon regulatory factor (IRF) pathways in response to pathogens. In contrast to the systemic response, severely ill COVID-19 patients typically experience a robust and prolonged inflammatory response in the lung compartment.

Regarding COVID-19 management, there is no prevailing breakthrough strategy that significantly differs (apart from the antimicrobials/antivirals) from the established sepsis treatment bundle recommended by the US National Institutes of Health guidelines. One important exception is the dissimilar efficacy of glucocorticoids (GC). While the current sepsis guidelines provide a weak recommendation for glucocorticoids, their use in severe SARS-CoV-2 pneumonia is unequivocally beneficial [15]. The biological mechanism responsible for this difference remains unclear and must be elucidated. Understanding the underlying reasons could potentially lead to a resurgence of GC use in bacterial sepsis and critical care in general. Similarly, some immune-therapies appear to confer amelioration for some COVID-19 patients [30,31], while this fact has not been proven for sepsis cases. As a result of these observations, the National Institutes of Health (NIH), EMA, and other international institutions issued a daily updated guideline that summarizes the recommended immunotherapies against COVID-19 and ongoing clinical trials (Table 2) [32].

Besides IL-6, our knowledge about the concentrations of other proinflammatory or anti-inflammatory mediators in patients with COVID-19 is still limited. Our understanding of the cytokine storm landscape, especially with regards to the chemokines that regulate the distribution and activity of effector cell populations, remains unclear. Interpreting changes in plasma cytokine concentrations that appear elevated without considering additional immune cellular parameters, does not provide clarity about the molecular basis of COVID-19 [33]. As a consequence, choosing an appropriate treatment strategy becomes a challenge.

IL-10, a pleiotropic cytokine known for its potent anti-inflammatory and immunosuppressive effects, has also been found to be elevated in COVID-19 patients [34]. This could lead to different conclusions regarding therapeutic approaches and our understanding of the disease’s pathophysiology [33]. However, the role of IL-10 is currently under re-evaluation. In addition to the classical IL-10 role as an anti-inflammatory cytokine, non-classical pro-inflammatory effects of IL-10 provide a plausible explanation for elevated IL-10 levels in the face of systemic inflammation [35].

Profound lymphopenia, an abnormally low level of lymphocytes in the blood similar to levels often seen in septic shock, is consistently observed in severely ill COVID-19 patients. This condition correlates with higher rates of secondary infections and mortality. The decrease and loss of immune effector cells is observed across all lymphocyte types, including CD8+ and natural killer cells, which have crucial antiviral roles, as well as B cells, which are essential for producing antibodies that neutralize the virus [33,36,37]. As a consequence, in addition to the “cytokine storm” hypothesis, another hypothesis has been suggested, namely that severe immunosuppression and not a cytokine storm characterizes COVID-19 infections [36]. The authors continue to suggest that treatments that support host protective immunity must be considered [22]. The most rational approach to support the patient’s protective immunity is to evaluate immune stimulants targeting the adaptive immunity and T-cell function [29,36,38]. Monoclonal antibody checkpoint inhibitors, nivolumab (Opdivom) and pembrolizumab (Keytruda) targeting PD-1, as well as IL-7 have been studied in clinical trials (Table 2) [39]. The inhibition of IL-7 has shown a beneficial effect in septic patients with an increase in the lymphocyte counts [40,41]. An aspect about the controversial two hypotheses is the current inability to address them due to a lack of appropriate diagnostic tools to evaluate cell immune function in COVID-19 infections [22].

Regardless of the differences with respect to immunotherapy, the importance of antimicrobial treatments and supportive therapies (e.g., oxygen for hypoxaemia and ventilatory support) are lessons learned from sepsis that can be transferred to COVID-19 patients. As in other infections leading to sepsis, successful treatment against COVID-19 involves eradication of the causative organism, namely SARS-CoV-2. Since the WHO declared the COVID-19 pandemic on March 2020 [42], scientists and clinicians around the world have worked around the clock to develop therapies, diagnostic kits, and vaccines against SARS-CoV-2. Many of those discoveries were first approved globally as temporary emergency use authorizations (EUA) by the Food and Drug Administration (FDA) in the USA and its international counterparts worldwide. As such, several EUAs were issued to treat COVID-19 that allowed the use of unapproved drugs or unapproved uses of approved drugs in the absence of alternatives. The European Medicines Agency (EMA) took a similar approach by granting conditional marketing authorization (CMA) to those types of drugs including both antivirals and antibodies. Some EUAs or CMA were later revised after some of the antibodies became ineffective against the Omicron variant of the virus (Table 4).

Additionally, the use of combination therapies has been proposed [43]. In this context, it was found that the antiviral activity of lactoferrin makes it a potential immunity enhancer which could be administered along with conventional antivirals [44]. Interestingly, this compound shows anti-SARS-CoV2 activity by itself [45], which seems to be mechanistically independent from its antibacterial and LPS-binding activities [46]. On the other hand, Sohn et al. (2020) [47] discovered that drugs that have been described as inhibitors of the LPS-induced cytokine storm such as the polypeptide Aspidasep (Pep19-2.5) [48,49,50,51,52] may also be useful against SARS-Cov2-induced hyperinflammation [47]. This may open the door to a new therapeutic approach against SARS-CoV-2.

## 3. Bacterial Coinfections and the Relationship between LPS and SARS-CoV-2

Bacterial coinfections with SARS-CoV-2 seem to be as prevalent as they once were with influenza virus from serotype H1N1, the etiological agent that caused the 1918 influenza pandemic, and they are believed to have played a significant role in the lethality of both diseases [53].

Bacterial coinfections or secondary bacterial infections are indeed critical risk factors for the severity and mortality rates of COVID-19 [54]. In addition, there is evidence supporting that most of the deaths during the 1918 influenza pandemic were due to the bacterial coinfections rather than the flu virus. In accordance with this hypothesis, serotype H1N1 influenza virus continues to have widespread prevalence worldwide without the devastating consequences of the 1918 pandemic. Nevertheless, there are many important differences between COVID-19 and the 1918 influenza pandemic. For instance, whereas the latter mainly affected young and fully immune-competent people, morbi-mortality due to COVID-19 was strongly associated with aging [55], comorbidities (see above), and immune deficiencies [56].

On the other hand, the cell mediators induced in the case of Gram-negative (lipopolysaccharide, LPS) [57], Gram-positive (lipoproteins or peptides) [58], and SARS-CoV-2 infections (see above, [57,59]) are remarkably similar. In this regard, it is worth noting that the most potent pathogen associated molecular patterns (PAMPs) from Gram-negative bacteria and SARS-CoV-2 induce inflammation through the same cell receptor, namely Toll-like receptor 4 (TLR4)/myeloid differentiation factor 2 (MD-2). Importantly, TLR4 is responsible for the intracellular cascade leading to pro-inflammatory cytokine secretion and its canonical agonist is LPS (endotoxin). Bacterial endotoxin ranks among the most potent immunostimulants found in nature and is the main triggering factor of Gram-negative sepsis, which affects millions of people worldwide [60].

In addition to well-known or presumed disorders triggered by bacteria such as colitis and Crohn’s disease, a variety of additional pathologies are due to the interaction of LPS with the immune system [61]. Such interactions can be the consequence of infections with Gram-negative bacteria, and/or be due to contact with commensal bacteria (Figure 1). The main concentration of this molecule is found in the gut that can contain up to 1.0 to 1.5 kg of bacteria [62]. However, there might also be significant concentrations in subepithelial tissues and in the liver [63]. 

Since mucosal barrier alterations may play a role in the development of several diseases, including COVID-19, Teixeira et al. (2021) examined the connection between bacterial translocation and systemic inflammation markers at the beginning of hospitalization (T1), and during the last 72 h (T2) in surviving and non-surviving COVID-19 patients. Blood samples were collected from 66 SARS-CoV-2 RT-PCR-positive patients and 9 non-COVID-19 pneumonia controls and the levels of systemic cytokines and chemokines, LPS concentrations, and soluble CD14 (sCD14) were analyzed by incubating the human THP-1 monocytic cell line with plasma from survivors and non-survivors. Their phenotype, activation status, TLR4, and chemokine receptors were analyzed by flow cytometry and confirmed that severe COVID-19 patients have increased LPS levels and systemic inflammation that result in monocyte activation [65].

Several animal models were developed to study COVID-19 infection and potential therapies. Since mouse ACE2 does not effectively bind the viral S protein, transgenic mouse models expressing human ACE2 were used [66]. These mice were susceptible to SARS-CoV-2 infection, but they differed in disease severity. More recently, new animal models have been created to faithfully replicate various aspects of COVID-19 in humans, with a specific focus on pulmonary vascular disease and ARDS [67]. For instance, a mouse inflammation model based on the coadministration of aerosolized SARS-CoV-2 S protein together with LPS to the lungs has been developed [68]. Using nuclear factor-kappa B (NF-κB) luciferase reporter and C57BL/6 mice followed by combinations of bioimaging, cytokine, chemokine, FACS, and histochemistry analyses, the model showed severe pulmonary inflammation and a cytokine profile similar to that observed in COVID-19. This animal model revealed a previously unknown high-affinity interaction between the SARS-CoV-2 S protein and LPS from *E. coli* and *P. aeruginosa*, leading to a hyperinflammation in vitro as well as in vivo [68]. Very importantly, the molecular mechanism underlying this effect was dependent on specific and distinct interactions between the S protein and LPS, enabling LPS’s transfer to CD14 and subsequent downstream NF-κB activation. The resulting synergism between the S protein and LPS has clinical relevance, providing new insights into comorbidities that may increase the risk for ARDS during COVID-19. In addition, microscale thermophoresis assays have yielded a KD of 47 nM for the interaction between LPS and SARS-CoV-2 S protein, slightly higher than the interaction between LPS and CD14 (45 nM). Computational modeling and all-atom molecular dynamics simulations further substantiated the experimental results, identifying a main LPS-binding site in SARS-CoV-2 S protein. S protein, when combined with low levels of LPS, boosted (NF-kB) activation in monocytic THP-1 cells and cytokine responses in human blood and peripheral blood mononuclear cells, respectively [63].

The data of the interaction of the S protein with LPS should be discussed in light of immune stimulation induced by LPS. There are various scenarios possible, and one hypothesis is that LPS is transferred to CD14 which then induces cell activation via the interaction of LPS with the complex of TLR4 and MD-2 [69]. A role of the LPS-binding protein LBP is also envisioned, although cell activation may also take place in the absence of LBP [70]. In any case, today it is assumed that for cell stimulation, the aggregate structure of LPS is decisive [71]. It has been shown that LPS monomers are biologically inactive [72]. LPS molecules naturally form aggregates that can lead to high activity when they are in a non-lamellar geometry, and display no activity in a lamellar form [73]. The different possible aggregate structures for LPS depend on the chemical structures of the monomers (Figure 2). In standard LPS, the lipid A part, the endotoxin principle, has a hexa-acylated diglucosamine backbone which is highly active. Other LPS that are under-acylated, for example with a tetra- or a penta-acylated lipid A, lack bioactivity [74,75,76]. In an analogy to this behavior, biologically active LPS converts, when it is inactivated by the addition of, for example, antimicrobial peptides such as compound Pep19-2.5 or polymyxin B, into a (multi)lamellar and thus, inactive aggregate [77,78].

From the foregoing, it is apparent that the binding of the S protein to LPS changes the conformation of the latter in a way that increases its stimulation potency. Therefore, an analysis of the S protein:LPS complex would give more insight for an understanding of the changes in bioactivity. Recently, biophysical investigations with the S protein have been performed [63,79]. Where a dynamic light scattering (DLS) assay showed that an incubation of SARS-CoV-2 S protein with either 100, 250, or 500 μg/mL of LPS yielded a significant reduction of the hydrodynamic radii of the LPS particles in solution, transmission electron microscopy (TEM) showed larger aggregates in the samples with 250 or 500 μg/mL of LPS. This was further confirmed by incubating a fixed concentration of LPS with either 5 nM of SARS-CoV-2 S protein that caused the disaggregation of LPS, or higher levels that induced its aggregation. These findings indicate that the interaction of S protein with LPS complexes is concentration-dependent, leading first to disaggregation and then again to an increase with corresponding differences in biological activity. For a biophysical understanding of these processes, analyses based on the methodology of the publications quoted in the legends of Figure 2 (e.g., small-angle X-ray scattering) would be necessary. 

According to the various papers cited above, it seems that LPS has a fundamental role in the expression of infectivity. In each case an enhancing action of LPS can be found. Interestingly, higher amounts of LPS and soluble CD14, a transport protein of LPS, was found in COVID-19 patients. Therefore, the question arises whether the infection caused by the SARS-Cov2 virus influences the metabolism of LPS in a way that leads to the observed detrimental effects of the infection.

The authors of [43,68,80] studied the coinfection of SARS-Cov2 with viruses, bacteria, and fungi, and discussed the reasons of the co-infection, their diagnosis, and their medical importance.

## 4. Influence of SARS-CoV-2 on the Coagulation System

Coagulopathy, with an incidence as high as 50% in patients with severe COVID-19, is frequent during both conventional sepsis and COVID-19. Coagulopathy in COVID-19 can be triggered by an increase in the vasoconstrictor angiotensin II, a decrease in the vasodilator angiotensin, and the sepsis-induced release of cytokines [81]. However, the effects of COVID-19 on the coagulation system are far from the typical disseminated intravascular coagulation (DIC) seen during bacterial sepsis [82]. While bacterial coagulopathy is associated with coagulation factor X, COVID-19-associated coagulopathy is characterized by elevated circulating fibrinogen, high levels of D-dimer, thrombocytopenia, and mildly affected clotting times [83]. In addition, pulmonary microvascular thrombosis has been reported and may play a role in progressive lung failure [84].

Unlike during conventional sepsis, anticoagulation seems to play a key role in the treatment of COVID-19. However, there is a lack of practice guidelines tailored to these patients. A scoring system for COVID-19-coagulopathy and stratification of patients for the purpose of anticoagulation therapy based on risk categories has been proposed [33]. In patients with shock, it was observed that antithrombin (AT) alone, but not the combined action of heparin and AT showed therapeutically favorable effects. Their proposed scoring system and therapeutic guidelines are likely to undergo revisions in the future as new data become available in this evolving field.

## 5. Long COVID-19 Syndrome

A notable similarity exists between bacterial sepsis and COVID-19 phenotypes: they both can cause long-term sequelae. In both patient groups, being discharged from the hospital does not equal a complete recovery, and it is instead often followed by prolonged, and debilitating consequences. While in bacterial sepsis, the post-discharge complications are referred to as post-sepsis syndrome or persistent inflammation, immunosuppression, and catabolism syndrome (PICS), in SARS-CoV-2 infected patients, these manifestations are known as “long COVID” [85,86]. According to the World Health Organization (WHO), “Long COVID” is defined as the continuation or development of new symptoms 3 months after the initial SARS-CoV-2 infection, with these symptoms lasting for at least 2 months with no other explanation. Although risk factors for long COVID include old age, female sex, and moderate or severe COVID-19, long COVID can develop regardless of disease severity [87].

The most common persistent symptoms for both long COVID and post-sepsis syndrome, include fatigue, muscle pain, poor sleep, and cardiac or cognitive disturbances (e.g., arrhythmias, short-term memory loss). Remarkably, a troubling difference exists between the two conditions. Unlike in post-sepsis syndrome, long-COVID is frequently diagnosed in mildly SARS-CoV-2-infected patients (i.e., those with no hospital stay). The presence of the “long-phenotype” in both illnesses strongly indicates a severe and prolonged deregulation of organ homeostasis and the immune–inflammatory system (with clear immunosuppression features). In the context of the slowly subsiding severe COVID-19 manifestations, one should re-focus on the long-term sequalae to evaluate a potential risk of increase in chronic debilitation.

## Figures and Tables

**Figure 1 ijms-24-15169-f001:**
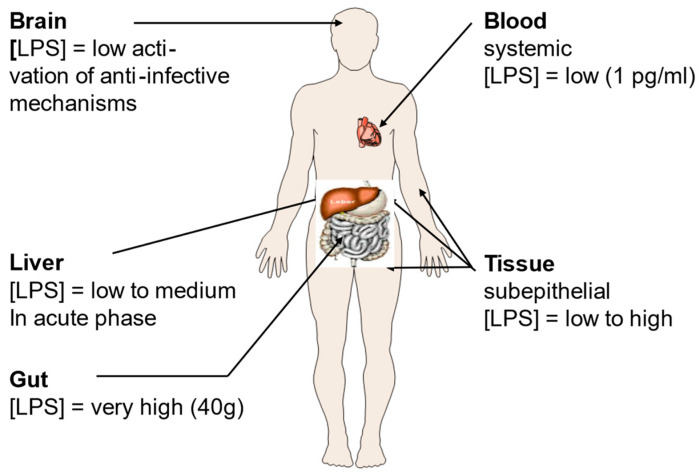
Lipopolysaccharide (LPS) concentrations in the human body. LPS is the main constituent of the outer leaflet of the outer membrane of Gram-negative bacteria, and it may induce inflammation even in the nanomolar range [64]. Its presence in the body is tightly associated with locations where bacteria are particularly abundant such as the gut, and the subepithelial tissue. Figure kindly provided by Robert Munford, Oxon, WA, USA.

**Figure 2 ijms-24-15169-f002:**
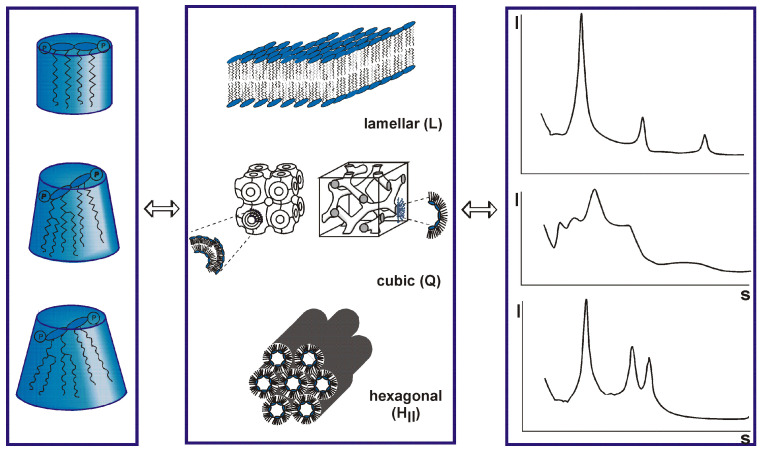
Varying conformations of lipopolysaccharide (LPS) monomer (left column) aggregates in different structures (middle panel). These different structures produce distinct small-angle X-ray patterns (right panel) and result from different degrees of acylation of the lipid A molecule (left panel). The acylation varies between tetra, (inactive, but antagonistic), penta (mostly inactive), hexa (normal form, highly active), and even hepta (similarly active as hexa). Unpublished results by K. Brandenburg et al. according to the theory of Israelachvili [73,74].

**Table 1 ijms-24-15169-t001:** Comorbidities of COVID-19 associated with disease severity. Data from non-redundant studies analyzed in references [5,6,7,8,9].

Risk Factor	Number of Studies	Total Sample Size	Association with COVID-19 Severity
Diabetes	142	59,476	Yes
Hypertension	140	58,808	Yes
Malignancy	94	48,488	Yes
Cerebrovascular disease	71	16,124	Yes
Chronic liver disease	56	27,924	Yes
COPD	50	32,173	Yes
Chronic kidney disease	43	20,103	Yes
Cardiovascular diseases	37	25,016	Yes
Coronary heart disease	33	16,525	Yes
Respiratory disease	31	7552	Yes
Chronic lung disease	31	3702	Yes
Chronic heart disease	9	3583	Yes
Autoimmune disease	7	2372	Yes
Renal insufficiency	6	2997	Yes
Stroke	5	1616	Yes
Cerebral infarction	4	2647	Yes
Fatty liver	4	992	Yes
Arrhythmia	4	781	Yes
Cardiac insufficiency	2	1912	Yes
Genital system diseases	2	546	Yes
Kidney failure	2	294	Yes
Coronary atherosclerosis	1	3044	Yes
Benign prostatic hyperplasia	1	3044	Yes
Myocardial infarction	1	660	Yes
Aorta sclerosis	1	140	No
Atrial fibrillation	1	112	No
Coronary artery disease	2	1073	No
Heart failure	1	172	No
Intracerebral hemorrhage	1	1767	No
Asthma	3	5359	No
Chronic bronchitis	2	2525	No
Tuberculosis	7	4125	No
Nephritis	1	3044	No
Gallbladder disease	3	779	No
Hepatitis B	6	3307	No
Gastrointestinal disease	6	4764	No
Peptic ulcer	1	145	No
Gout	1	134	No
Hyperlipidemia	7	4131	No
Hyperuricemia	1	172	No
Thyroid disease	5	1125	No
Cirrhosis	3	5134	No
Prostatitis	1	3044	No
Gynecological disease	1	238	No
HIV infection	7	1099	No
Nervous system disease	5	2203	No
Rheumatism	2	273	No
Urinary system disease	2	1075	No
Urolithiasis	1	140	No
Blood system diseases	3	965	No
Bone disease	1	238	No

**Table 2 ijms-24-15169-t002:** Immunotherapies against COVID-19.

Mechanism	Drug Family	Drugs	Status
Anti-inflammatory drugs	Systemic glucocorticoids	Dexamethasone, Prednisone, Hydrocortisone, Methylprednisolone	Recommended for certain hospitalized patients
	Anti-IL-6 receptor antibodies	Tocilizumab, Sarulimab	Recommended for certain hospitalized patients
	Anti-IL-6 antibody	Siltuximab	Not recommended. Under investigation in clinical trials
	IL-1 receptor antagonists	Anakinra, Canakinumab	Anakinra received an FDA EUA for certain hospitalized patients. Canakinumab is not recommended
	JAK/STAT inhibitors	Baricitinib, Tofacitinib, Ruxolitinib	Baricitinib and Tofacitinib recommended for certain hospitalized patients. Ruxolitinib under investigation in clinical trials
	GM-CSF inhibitors	Lenzilumab, Mavrilimumab, Namilumab, Otilimab, Gimsilumab	Not recommended. Under investigation in clinical trials
	TNF-alpha inhibitor	XPro1595, CERC-002, Infliximab, Adalimumab	Not recommended. Under investigation in clinical trials
Immune stimulants	Programmed death ligand pathway inhibitors	Nivolumab and Pembrolizumab	Not recommended. Under investigation in clinical trials
	IL-7		Not recommended. Under investigation in clinical trials
	IFN-γ		Not recommended. Under investigation in clinical trials
	NKG2D-ACE2 CAR-NK cells		Not recommended. Under investigation in clinical trials

**Table 3 ijms-24-15169-t003:** Comparison of sepsis and COVID-19: disease evolution [29].

	Early Sepsis	Early COVID-19	Late Sepsis	Late COVID-19
IL-6 increase	+++	+		+++
Lymphopenia	+	++	++	+++
Nosocomial infections			+++	++

**Table 4 ijms-24-15169-t004:** Antivirals and antibodies granted an emergency use authorization (EUA) by the FDA or a conditional marketing authorization (CMA) by the European Medicines Agency (EMA) during COVID-19 pandemic.

	Drug	Brand Name	FDA EUA	EMA CMA	Rescinded-Revised by FDA/EMA
Antivirals	Hydroxychloroquine sulfate Chloroquine phosphate	Several	March 2020		June 2020
	Remdesivir	Veklury	May 2020	June 2020	
	Nirmatrelvir/Ritonavir	Paxlovid	December 2021	January 2022	
	Molnupiravir	Lagevrio	December 2021		
Anti-SARS-CoV-2-antibodies	Convalescent plasma		August 2020		
	Bamlanivimab		November 2020	March 2021	January 2022/ November 2021
	Casirivimab/ Imdevimab	Regen-cov2	November 2020	February 2021	January 2022
	Etesevimab		December 2021	March 2021	January 2022/ November 2021
	Tixagevimab/ Cilgavimab	Evusheld	December 2021	March 2022	
	Sotrovimab	Xevudy	January 2022	May 2021	
	Regdanvimab	Regkirona		November 2021	

## Data Availability

Data on X-ray scattering of bacterial toxins and on the biological activities of the antimicrobial peptide Aspidasept (Pep19-2.5) can be obtained from “Brandenburg Antiinfectiva GmbH” (K.B. and K.M.).

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
