# Peer review of "A Comparison between SARS-CoV-2 and Gram-Negative Bacteria-Induced Hyperinflammation and Sepsis"

_ijms, 2023, doi:10.3390/ijms242015169_

Round 1

Reviewer 1 Report

The manuscript by Brandenburg et al., “A comparison between SARS-CoV2 and Gram-negative bacteria-induced hyperinflammation and sepsis” describes a critical look at the patterns that occur in both sepsis and COVID-19 in the light of the immune responses and the possible mechanisms involved in the observed symptoms.

I have carefully read through the manuscript and have these few comments to make to improve on understanding and add clarity.

1.     The authors write about COVID-19 in the past whereas the pandemic is still ongoing. I, therefore, see a need to change the tense in which the abstract and some parts of the manuscript are written. In other places, there is a mix of the past and present tenses.

2.     References are missing for some sentences in many sections of the manuscript e.g., lines 112-124, 200 – 203, 354-355. In some cases, the references are not clear, e.g., Ahlstrom et al., line 68, with references, 8,9, and 10. Also, I think the authors should use a referencing style that references different sentences separately and not blocks of sentences.

3.     In line 64, the authors begin a sentence with, “This meta-analysis…..” without laying down a proper context as to what they are referring to. Similarly, in line 161, another sentence begins with, “The authors…..”. These should be checked again and corrected.

4.     In Table 4, the authors should check the spelling for the months of the year.

5.     Scientific names should be italicized

6.     Lastly, the authors should be consistent in the manner of writing COVID-19. Sometimes, it is written as Covid-19

The mix of tenses needs to be corrected (point 1 above)

Author Response

The authors of “A comparison between SARS-CoV-2 and Gram-negative bacteria-induced hyperinflammation and sepsis” thank the reviewers and the journal their efforts on improving the quality of the present review. We have taken their comments seriously and implemented all their requests. This is a summary of the changes undertaken that can also be seen in yellow in the manuscript:

REVIEWER 1

  1. The authors write about COVID-19 in the past whereas the pandemic is still ongoing. I, therefore, see a need to change the tense in which the abstract and some parts of the manuscript are written. In other places, there is a mix of the past and present tenses.

The tenses have been changed to the present tense. See for example lines 28, and 44.

  1. References are missing for some sentences in many sections of the manuscript e.g., lines 112-124, 200 – 203, 354-355. In some cases, the references are not clear, e.g., Ahlstrom et al., line 68, with references, 8,9, and 10. Also, I think the authors should use a referencing style that references different sentences separately and not blocks of sentences.

References have been added:

Previous line

Current line

Sentence

Reference added

112-124

120

According to several studies, the inflammatory phase for patients with severe COVID-19 is limited to the initial period of the disease.

28

200 – 203

210

Bacterial coinfections with SARS-CoV-2 seem to be as prevalent as they once were with influenza virus from serotype H1N1, the etiological agent that caused the 1918 influenza pandemic, and they are believed to have played a significant role in the lethality of both diseases.

53

354-355

339

In addition, pulmonary microvascular thrombosis has been reported and may play a role in progressive lung failure.

84

68

76

To address this hypothesis, Ahlström et al compared the impact of comorbidities on mortality in patients with COVID‑19 and sepsis

Has been reduced to reference 8

  1. In line 64, the authors begin a sentence with, “This meta-analysis.....” without laying down a proper context as to what they are referring to. Similarly, in line 161, another sentence begins with, “The authors.....”. These should be checked again and corrected.

Sentences have been clarified and references have been added.

Previous line

Current line

Sentence

Reference added

161

170

The authors continue to suggest that treatments that support host protective immunity must be considered

22

64

69

Chen et al reported that acute respiratory distress syndrome (ARDS), respiratory failure, multiple organ dysfunction syndrome (MODS) and septic shock were complications strongly associated with critical cases of coronavirus disease[5]. This meta-analysis was particularly. relevant as it examined data from 187 studies describing 77,013 patients[5]

We have added the name of the study in the previous sentence and cited it on both sentences.

5

  1. In Table 4, the authors should check the spelling for the months of the year.

The names of the months on table 4 have been corrected.

  1. Scientific names should be italicized.

Scientific names have been italicized. See line 270.

  1. Lastly, the authors should be consistent in the manner of writing COVID-19. Sometimes, it is written as Covid-19.

Covid-19 has been corrected to COVID-19. See for example line 61.

  1. The mix of tenses needs to be corrected (point 1 above)

Tenses have been addressed.

Reviewer 2 Report

This manuscript shows rich content, providing a deep insight for some works: the study is within the journal’s scope, and I found it to be well-written, providing sufficient information. Even if the manuscript provides an organic overview, with a densely organized structure and based on well-synthetized evidence, there are some suggestions necessary to make the article complete and fully readable. For these reasons, the manuscript requires major changes.

Please find below an enumerated list of comments on my review of the manuscript:

The authors should provide a list of the abbreviations, mentioned in this manuscript.

ABSTRACT:

The authors should express the main aim of the manuscript in an organic manner.

INTRODUCTION:

LINE 46: The causative agent for COVID-19, severe acute respiratory syndrome coronavirus-2 (SARS-CoV-2), is an enveloped positive single-stranded RNA virus, with the most prominent viral genome of 8.4–12 kDa in size. There is a 5’ terminal in this viral genome, the central part of this genome, rich in open reading frames, which encodes proteins essential for virus replication. Instead, the 3’ terminal includes five structural proteins, Spike protein (S), responsible for the pathogenesis in the human species, the membrane protein (M), nucleocapsid protein (N), envelope protein (E), and hemagglutinin-esterase protein (HE) (see, for reference: https://doi.org/10.3390/pathogens11080867). In this introductive section, the authors should provide recent evidence on the molecular dynamics which underlie SARS-CoV-2 infection: according to the journal’s scope, the manuscript may benefit from describing the most significant molecular features of the causative agent of COVID-19.

LINE 365: According to the World Health Organization (WHO), “Long COVID” is defined as the continuation or development of new symptoms 3 months after the initial SARS-CoV-2 infection, with these symptoms lasting for at least 2 months with no other explanation. Although risk factors for long COVID include old age, female sex, and moderate or severe COVID-19, long COVID can develop regardless of disease severity (see, for reference: https://doi.org/10.1038%2Fs41598-023-36995-4). Before describing the parallelism between bacterial sepsis and COVID-19, the manuscript may benefit from providing an organic description of Long COVID, according to WHO and recent evidence on this topic.

The hallmark of this manuscript is the great clinical impact. This review in fact is interesting, original and a significant contribute to the ongoing research on this topic. There are rich contents and intersting insights on previous and contemporary works. Furthermore, there is a specific and detailed explanation for the evidence mentioned in this study: this is particularly significant, since the manuscript relies on a multitude of studies, to derive its conclusions. The methodology applied is overall correct, the results are reliable and adequately discussed.

The conclusion of this manuscript is perfectly in line with the main purpose of the paper: the authors have designed and conducted the study properly.

Finally, this manuscript also shows a basic structure, properly divided and looks like very informative on this topic. Furthermore, figures and tables are complete, organized in an organic manner and easy to read.

In conclusion, this manuscript is densely presented and well organized, based on well-synthetized evidence. This manuscript provided a comprehensive analysis of current knowledge in this field. Moreover, this research has futuristic importance and could be potential for future research. However, major concerns of this manuscript are with the introductive and discussive sections: for these reasons, I have major comments for these sections, for improvement before acceptance for publication. The article is accurate and provides relevant information on the topic and I have some major points to make, that may help to improve the quality of the current manuscript and maximize its scientific impact. I would accept this manuscript if the comments are addressed properly.

Minor editing of English Language are necessary.

Author Response

The authors of “A comparison between SARS-CoV-2 and Gram-negative bacteria-induced hyperinflammation and sepsis” thank the reviewers and the journal their efforts on improving the quality of the present review. We have taken their comments seriously and implemented all their requests. This is a summary of the changes undertaken that can also be seen in yellow in the manuscript:

REVIEWER 2

  1. The authors should provide a list of the abbreviations, mentioned in this manuscript.

A list of the abbreviations used has been added at the end of the manuscript (Line 622).

  1. The authors should express the main aim of the manuscript in an organic manner.

We have added a line at the end of the abstract (Line 38).

  1. LINE 46: The causative agent for COVID-19, severe acute respiratory syndrome coronavirus-2 (SARS-CoV-2), is an enveloped positive single-stranded RNA virus, with the most prominent viral genome of 8.4–12 kDa in size. There is a 5’ terminal in this viral genome, the central part of this genome, rich in open reading frames, which encodes proteins essential for virus replication. Instead, the 3’ terminal includes five structural proteins, Spike protein (S), responsible for the pathogenesis in the human species, the membrane protein (M), nucleocapsid protein (N), envelope protein (E), and hemagglutinin-esterase protein (HE) (see, for reference: https://doi.org/10.3390/pathogens11080867). In this introductive section, the authors should provide recent evidence on the molecular dynamics which underlie SARS-CoV-2 infection: according to the journal’s scope, the manuscript may benefit from describing the most significant molecular features of the causative agent of COVID-19.

The suggested information has been added in Line 53.

  1. LINE 365: According to the World Health Organization (WHO), “Long COVID” is defined as the continuation or development of new symptoms 3 months after the initial SARS-CoV-2 infection, with these symptoms lasting for at least 2 months with no other explanation. Although risk factors for long COVID include old age, female sex, and moderate or severe COVID-19, long COVID can develop regardless of disease severity (see, for reference: https://doi.org/10.1038%2Fs41598- 023-36995-4). Before describing the parallelism between bacterial sepsis and COVID-19, the manuscript may benefit from providing an organic description of Long COVID, according to WHO and recent evidence on this topic.

The suggested information has been added in Line 356.

JOURNAL TASKS

  1. Reduce self-citation.

We have removed the following references:

Brandenburg, K.; Wiese, A. Endotoxins: Relationships between Structure, Function, and Activity. Current Topics in Medicinal Chemistry 2004, 4, 1127–1146.

Gutsmann, T.; Howe, J.; Zähringer, U.; Garidel, P.; Schromm, A.B.; Koch, M.H.J.; Fujimoto, Y.; Fukase, K.; Moriyon, I.; Martínez-de-Tejada, G.; et al. Structural Prerequisites for Endotoxic Activity in the Limulus Test as Compared to Cytokine Production in Mononuclear Cells. Innate Immun 2010, 16, 39–47, doi:10.1177/1753425909106447.

Brandenburg, K.; David, A.; Howe, J.; Koch, M.H.J.; Andrä, J.; Garidel, P. Temperature Dependence of the Binding of Endotoxins to the Polycationic Peptides Polymyxin B and Its Nonapeptide. Biophysical Journal 2005, 88, 1845–1858, doi:10.1529/biophysj.104.047944.

  1. Request permission for references in Figure 2 (or replace those references with other more suitable).

We have removed reference:

Brandenburg, K.; Wiese, A. Endotoxins: Relationships between Structure, Function, and Activity. Current Topics in Medicinal Chemistry 2004, 4, 1127–1146.

And we have left reference:

Israelachvili, J.N. Chapter 19: Thermodynamic Principles of Self-Assembly. In Intermolecular & Surface Forces; Academic Press Ltd: London, San Diego, New York, Boston, Sydney, Tokyo, Toronto, 1991; Vol. 2, pp. 341–365 ISBN 978-0-12-375182-9.

This figure is unpublished results by K. Brandenburg et al, according to the theory of J. Israelachvili. Which is a widely known and referenced book in the field.

  1. High repetition rate

We have re-written the sentences with numbers 1, 2 and 3 in the version sent by the journal.

Round 2

Reviewer 2 Report

The authors have significantly improved the quality of this manuscript.